# Changes in household food and drink purchases following restrictions on the advertisement of high fat, salt, and sugar products across the Transport for London network: A controlled interrupted time series analysis

Amy Yau[1,2]*, Nicolas Berger[1,3], Cherry Law[1], Laura Cornelsen[1], Robert Greener[1], Jean Adams[4], Emma J. Boyland[5], Thomas Burgoine[4], Frank de Vocht[6,7], Matt Egan[8], Vanessa Er[1,2], Amelia A. Lake[9,10], Karen Lock[2], Oliver Mytton[4], Mark Petticrew[8], Claire Thompson[11], Martin White[4], Steven Cummins[1]*

**1** Population Health Innovation Lab, Department of Public Health, Environments and Society, Faculty of Public Health and Policy, London School of Hygiene & Tropical Medicine, London, United Kingdom, **2** Department of Health Services Research and Policy, Faculty of Public Health and Policy, London School of Hygiene & Tropical Medicine, London, United Kingdom, **3** Department of Epidemiology and Public Health, Scientific Institute of Public Health (Sciensano), Brussels, Belgium, **4** MRC Epidemiology Unit, University of Cambridge, Cambridge, United Kingdom, **5** Department of Psychology, Institute of Population Health, University of Liverpool, Liverpool, United Kingdom, **6** Population Health Sciences, Bristol Medical School, University of Bristol, Bristol, United Kingdom, **7** National Institute for Health Research Applied Research Collaboration West, Bristol, United Kingdom, **8** Department of Public Health, Environments and Society, Faculty of Public Health and Policy, London School of Hygiene & Tropical Medicine, London, United Kingdom, **9** School of Health and Life Sciences, Teesside University, Middlesbrough, United Kingdom, **10** Fuse–The Centre for Translational Research in Public Health, Newcastle upon Tyne, United Kingdom, **11** Centre for Research in Public Health and Community Care, School of Health and Social Work, University of Hertfordshire, Hatfield, United Kingdom

* amy.yau@lshtm.ac.uk (AY); steven.cummins@lshtm.ac.uk (SC)

**Data Availability Statement:** Purchase data, nutrient data and social and demographic data

## Abstract

### Background

Restricting the advertisement of products with high fat, salt, and sugar (HFSS) content has been recommended as a policy tool to improve diet and tackle obesity, but the impact on HFSS purchasing is unknown. This study aimed to evaluate the impact of HFSS advertising restrictions, implemented across the London (UK) transport network in February 2019, on HFSS purchases.

### Methods and findings

Over 5 million take-home food and drink purchases were recorded by 1,970 households (London [intervention], $n = 977$; North of England [control], $n = 993$) randomly selected from the Kantar Fast Moving Consumer Goods panel. The intervention and control samples were similar in household characteristics but had small differences in main food shopper sex,

were provided by Kantar Worldpanel Plus. The terms of our data agreement with Kantar mean that we cannot share these data.

**Funding:** This study is funded by the National Institute for Health Research (NIHR) School for Public Health Research (grant number PD-SPH-2015). The NIHR School for Public Health Research is a partnership between the Universities of Sheffield; Bristol; Cambridge; Imperial; and University College London; The London School for Hygiene and Tropical Medicine (LSHTM); LiLaC – a collaboration between the Universities of Liverpool and Lancaster; and Fuse - The Centre for Translational Research in Public Health, a collaboration between Newcastle, Durham, Northumbria, Sunderland and Teesside Universities. SC is also funded by Health Data Research UK (HDR-UK). HDR-UK is an initiative funded by the UK Research and Innovation, Department of Health and Social Care (England) and the devolved administrations, and leading medical research charities. JA, TB and MW are supported by the MRC Epidemiology Unit, University of Cambridge [grant number MC/UU/12015/6] and Centre for Diet and Activity Research (CEDAR), a UK Clinical Research Collaboration (UKCRC) Public Health Research Centre of Excellence. Funding for CEDAR from the British Heart Foundation, Cancer Research UK, Economic and Social Research Council, Medical Research Council, the NIHR [grant numbers ES/G007462/1 and MR/K023187/1], and the Wellcome Trust [grant number 087636/Z/08/Z], under the auspices of the UKCRC, is gratefully acknowledged. CT is funded by the NIHR Applied Research Collaboration East of England. AAL is a member of Fuse, the Centre for Translational Research in Public Health (www.fuse.ac.uk). Fuse is a Public Health Research Centre of Excellence funded by the five North East Universities of Durham, Newcastle, Northumbria, Sunderland and Teesside. FdV is partly funded by the NIHR Applied Research Collaboration West (NIHR ARC West) at University Hospitals Bristol NHS Foundation Trust. LC is funded by a UK Medical Research Council fellowship (grant number MR/P021999/1). RG is funded by a UK Medical Research Council studentship [grant number MR/N013638/1]. The views expressed are those of the authors and do not necessarily represent those of any of the above named funders. The funders had no role in study design, data collection and analysis, decision to publish, or preparation of the manuscript.

**Competing interests:** I have read the journal's policy and the authors of this manuscript have the

socioeconomic position, and body mass index. Using a controlled interrupted time series design, we estimated average weekly household purchases of energy and nutrients from HFSS products in the post-intervention period (44 weeks) compared to a counterfactual constructed from the control and pre-intervention (36 weeks) series. Energy purchased from HFSS products was 6.7% (1,001.0 kcal, 95% CI 456.0 to 1,546.0) lower among intervention households compared to the counterfactual. Relative reductions in purchases of fat (57.9 g, 95% CI 22.1 to 93.7), saturated fat (26.4 g, 95% CI 12.4 to 40.4), and sugar (80.7 g, 95% CI 41.4 to 120.1) from HFSS products were also observed. Energy from chocolate and confectionery purchases was 19.4% (317.9 kcal, 95% CI 200.0 to 435.8) lower among intervention households than for the counterfactual, with corresponding relative reductions in fat (13.1 g, 95% CI 7.5 to 18.8), saturated fat (8.7 g, 95% CI 5.7 to 11.7), sugar (41.4 g, 95% CI 27.4 to 55.4), and salt (0.2 g, 95% CI 0.1 to 0.2) purchased from chocolate and confectionery. Relative reductions are in the context of secular increases in HFSS purchases in both the intervention and control areas, so the policy was associated with attenuated growth of HFSS purchases rather than absolute reduction in HFSS purchases. Study limitations include the lack of out-of-home purchases in our analyses and not being able to assess the sustainability of observed changes beyond 44 weeks.

## Conclusions

This study finds an association between the implementation of restrictions on outdoor HFSS advertising and relative reductions in energy, sugar, and fat purchased from HFSS products. These findings provide support for policies that restrict HFSS advertising as a tool to reduce purchases of HFSS products.

## Author summary

### Why was this study done?

- Many governments and local authorities are considering advertising restrictions to reduce consumption of high fat, salt, and sugar (HFSS) products as part of obesity prevention strategies.

- Evidence of the effectiveness of such policies in reducing purchases of HFSS products is limited, especially outside of broadcast media.

- The introduction of an outdoor advertising policy across a large transport network provided an opportunity to evaluate a natural experiment assessing whether implementation of such a policy is associated with changes in household food and drink purchases.

### What did the researchers do and find?

- We compared average weekly household purchases of HFSS products by households in the intervention area ($n = 977$) to an estimation of what would have happened without the policy—a counterfactual scenario estimated by extrapolating the pre-

following competing interests: JA is an Academic Editor for PLOS Medicine.

**Abbreviations:** BMI, body mass index; CITS, controlled interrupted time series; FMCG, Fast Moving Consumer Goods; HFSS, high fat, salt, and sugar; NPM, Nutrient Profiling Model; SDIL, Soft Drinks Industry Levy; TfL, Transport for London.

implementation trend and accounting for the post-implementation changes seen in households in the control area ($n = 993$).

- The average weekly household purchase of energy from HFSS products was 6.7% (1,001.0 kcal) lower in intervention households after the introduction of the policy compared to the counterfactual, and 19.4% (317.9 kcal) lower for energy from chocolate and confectionery.

- Average weekly household purchases of nutrients from HFSS products were lower after intervention implementation in intervention households relative to the counterfactual: fat, by 57.9 g; saturated fat, by 26.4 g; and sugar, by 80.7 g.

## What do these findings mean?

- Our findings provide support for advertising restrictions as a tool to reduce purchases of energy, sugar, and fat from HFSS products.

- This study adds to the evidence that can be used by governments and local authorities when developing obesity prevention strategies.

## Introduction

The advertisement of foods and drinks with high fat, salt, and sugar (HFSS) content is known to be associated with poor diet and obesity, particularly in children [1–4]. There is a high prevalence of exposure to HFSS food and drink advertising across a variety of media, especially among disadvantaged groups and in more deprived areas [5–9]. In children, exposure to advertising of HFSS foods and drinks has been associated with preferences for HFSS products, requests for purchases, and higher consumption of HFSS products [1,2,4]. Associations in adults are not as well-studied, and findings are inconsistent. However, some studies have found exposure to advertising of HFSS foods and drinks to be positively associated with purchasing and consumption of HFSS products and body mass index (BMI) [10–14]. Evidence suggests that advertisement of HFSS foods and drinks can influence food behaviours by changing dietary norms and shifting population-level food and drink preferences [15].

Policies that restrict the advertising of HFSS products have been promoted as potentially effective tools to reduce the purchase and consumption of HFSS products, with the aim of improving diet, reducing obesity and diet-related diseases, and tackling health inequalities [16,17]. Policies on HFSS advertising have been implemented in many countries, but most policies have been limited in scope, with a focus on broadcast advertising and a reliance on voluntary agreements by the food and advertising industries [18,19]. These voluntary commitments have been limited in effectiveness [19–21]. In 2007, the United Kingdom (UK) implemented statutory regulations to limit children's exposure to HFSS food and drink advertising through television [22]. Ofcom, the UK's communications regulator, estimated that children saw 34% fewer HFSS advertisements as a result [23]. However, as the regulations only applied to programmes targeting children, evidence suggests that there was displacement of advertising to television programmes aimed at a mixed audience. Independent research estimated that the policy had no effect on children's overall exposure to HFSS advertising and that it increased

exposure for the population as a whole [24]. Studies of advertising restrictions from other countries and regions provide evidence of statutory regulations reducing the volume of, or exposure to, HFSS advertising [21,25,26] and reducing purchases of HFSS products [19,27]. However, limited evidence exists on the effectiveness of policies that restrict advertising of HFSS products outside of broadcast media [18,20,28]. One study evaluating restrictions on fast food advertising across multiple print and electronic media in Quebec, Canada, found a 13% decrease in likelihood of purchasing following the introduction of the policy [27].

In November 2018, restrictions on the outdoor advertising of HFSS foods and drinks across the Transport for London (TfL) network were announced by the Mayor of London, UK (Box 1) [29,30]. The TfL advertising estate includes the London Underground (rapid transit) network, the TfL Rail network, transport vehicles run by TfL (including some buses, trains,

## Box 1. HFSS advertising restriction policy across the TfL estate

### Advertising and TfL

- Outdoor, or out-of-home, advertising is advertising seen in public spaces (e.g., billboards, on buses), often on the go and including digital and interactive displays [69].

- Outdoor advertising is thought to reach 98% of the UK population at least once a week and is particularly effective at reaching young, urban, affluent consumers [69].

- The TfL network is the largest transport network in Western Europe.

- TfL has one of the largest outdoor advertising estates in the UK, representing £152.1 million in advertising spending in 2017/2018 [29].

- The TfL estate accounts for 40% of London's outdoor advertising spending and 20% of outdoor advertising spending across the UK [29].

### TfL advertising policy

- Food and drink products are subject to advertising restrictions if they are classified as HFSS by the Nutrient Profiling Model (NPM) developed by the UK Food Standards Agency [37].

- Both direct marketing of HFSS products and incidental images of HFSS products in advertisements are prohibited [31].

- Companies can apply to advertise a product through an exceptions process if they can demonstrate that the product is not consumed by children and does not contribute to childhood obesity [31].

- Advertisements for some food categories (e.g., chocolate and confectionery) and for brands that do not produce non-HFSS products as part of their range will be completely removed from the TfL network.

- Brands with more diverse product ranges (e.g., drinks companies) may have low or zero calorie alternatives that allow continued brand advertising.

and taxis), and outdoor spaces owned by TfL (e.g., bus stops and land outside train stations) [29]. The restrictions were fully implemented on 25 February 2019. A full description of the policy guidance is available from the TfL website [31]. Though these restrictions formed part of a childhood obesity strategy, they may impact food behaviours (such as purchasing and consumption) across the whole population. We hypothesised that this policy may contribute to improvements in diet by reducing energy and nutrients purchased from HFSS products. In the absence of longitudinal dietary data, we used household purchase data to evaluate the impact of the intervention. This study aimed to estimate the changes in household purchases of energy and nutrients from HFSS products associated with the TfL advertising policy. Research investigating the implementation process [32] and the media representation of opposition to the policy [33] has been published.

## Methods

Using a controlled interrupted time series (CITS) analysis design, we estimated mean weekly household purchases of energy and nutrients from HFSS products, and packs of HFSS products, in London in the post-intervention period, compared with a counterfactual scenario where the intervention had not occurred. We examined purchases of all products classified as HFSS and within 5 key HFSS categories: (1) chocolate and confectionery, (2) puddings and biscuits, (3) sugary drinks, (4) sugary cereals, and (5) savoury snacks. Assignment to these categories was informed by previous work and is detailed in S1 Table [34]. The protocol was registered (ISRCTN 19928803) (S1 Text), and the study is reported in accordance with STROBE guidelines (for checklist, see S2 Text).

### Data

**Households.** Take-home grocery food and drink purchase data were available from households in the Kantar Fast Moving Consumer Goods (FMCG) panel from 18 June 2018 to 29 December 2019 (80 weeks), with 36 pre-intervention weeks and 44 post-intervention weeks. Kantar (a commercial consumer data company) recruits households to a live panel via email or post using quota sampling, and maintains a nationally representative sample of approximately 32,000 households annually. For this study, households ($n$ = 2,118) were randomly sampled for inclusion from London (intervention group) and the North of England (control group), based on postcode of residence. The North of England sample consisted of households in the North West, North East, and Yorkshire and the Humber regions (Fig 1). The North of England was chosen as a location-based control group due to its distance from London, reducing the likelihood of spillover effects (e.g., contamination of the control group through regular commuting to London from neighbouring counties) [35]. The control group enabled adjustment for the confounding effects that were common to both areas, including seasonal fluctuations and underlying trends in HFSS purchasing. Households recruited to the panel after the intervention was introduced (25 February 2019) were excluded from analyses ($n$ = 148) (Fig 2). Our analytical sample ($n$ = 1,970) included 977 intervention households and 993 control households.

**Food and drink purchases.** Participating households record all grocery (food and drink) items purchased and brought into the home, using a handheld barcode scanner [34]. Non-barcoded products, such as loose fruits and vegetables, are recorded using bespoke barcodes. In this study, 5,089,988 packs of 95,413 unique food and drink products were purchased. A pack is the individual item scanned by the participating household, which could be a single serving or a multipack, and therefore does not reflect volume purchased. For example, 23,564 (9.0%)

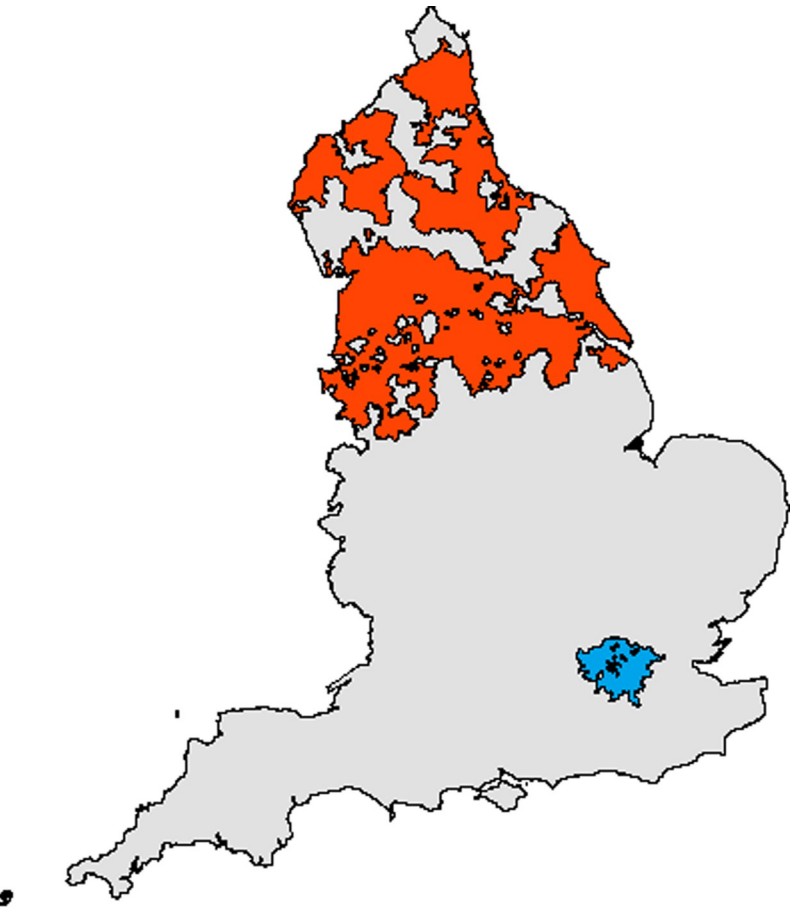

**Fig 1. Map of England showing the intervention (London) and control (North of England) areas.** Blue = London; red = North of England. This figure was created using a base map downloaded from https://osdatahub.os.uk/downloads/open/BoundaryLine.

chocolate and confectionery purchases were boxes of chocolates and 90,694 (34.7%) were multipacks, with a mean energy of 786.5 kcal per pack.

Data were aggregated to weekly purchases per household ($n = 157,600$). Most households did not report in all 80 study weeks (mean 70.7 [SD 8.7]; median 73 [IQR 66–78]). The reasons for non-reporting are unknown, but could include households habitually purchasing groceries less than weekly, going on holiday, or forgetting to report. The proportion of households with no reported purchases in a given week fluctuated, but with no clear pattern, so we assumed missingness to be random. Household-week observations where households did not report any food and drink purchases were dropped ($n = 18,407$ [11.7%]), resulting in 139,193 household-week observations for each outcome.

**Nutrients.** Nutritional data are collected by Kantar either through direct measurement in outlets twice a year or use of product images provided by Brandbank. Regular data collection helps to capture product reformulation. Where nutritional data cannot be collected directly, either nutritional values are copied across from similar products or an average value for the category or product type is calculated. The proportion of imputed values in our dataset is unknown. However, a previous study reported imputed values for between 11.0% (energy) and 19.6% (fibre) of the nutrient data from Kantar [34].

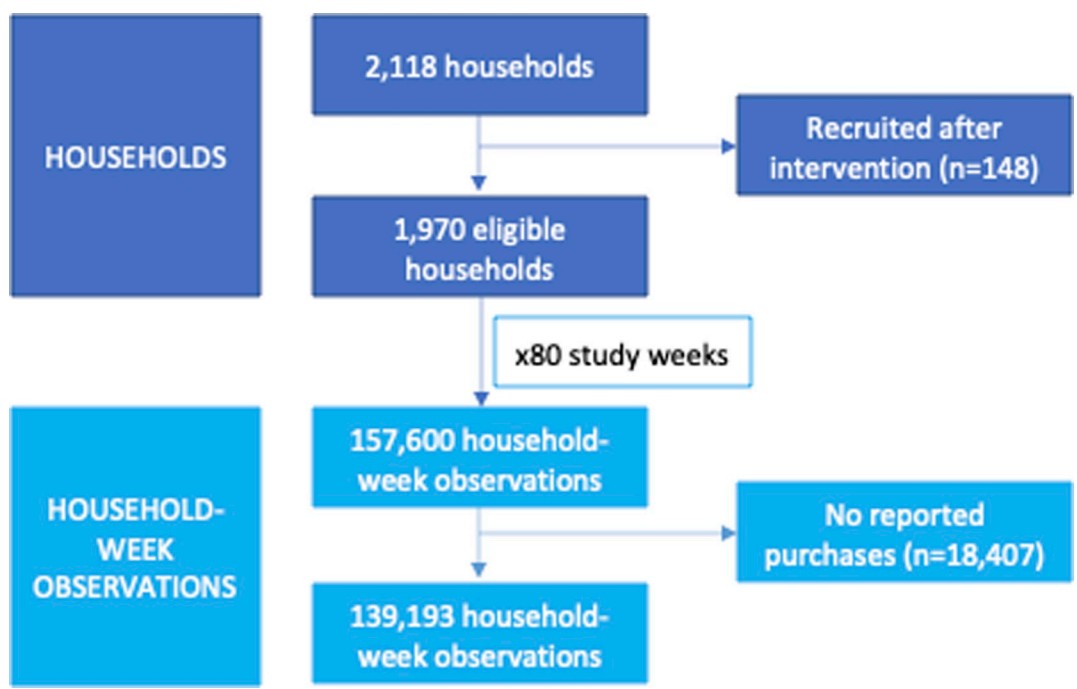

**Fig 2. Eligibility and inclusion of households and household-week observations.**

## Household and main food shopper characteristics

Sociodemographic data from the panellists are collected annually and include characteristics of the main food shopper: sex (male and female), age (years), socioeconomic position, household size, and weight status. Socioeconomic position is classified according to the National Readership Survey (NRS) occupational social grade classification (A, B, C1, C2, D, E) [36]. We categorised NRS social grade into 3 socioeconomic groups: high (AB), middle (C1C2), and low (DE). Data on household size included number of children <16 years and number of adults. The household main food shopper reports their weight and height (available for $n = 1,591$ [80.8%]), from which BMI (weight [kg]/height$^2$ [m$^2$]) is calculated. We classified main food shoppers as living with overweight or obesity (BMI $\geq$ 25 kg/m$^2$) or not (BMI < 25 kg/m$^2$). The main food shoppers from 1,296 (65.8%) households completed an additional survey in February 2019 in which they were asked about typical frequency of public transport use per week [8]. Households were then categorised as typically using public transport at least once a week (yes or no) for use in sensitivity analyses.

## Outcomes

We categorised products as HFSS (yes or no) according to the NPM, which was used to determine whether products could be advertised on the TfL estate [31,37]. An NPM score was calculated using points for energy, sugar, sodium, and saturated fat minus points for protein, fibre, and fruit and vegetable content. Information on the energy, sugar, sodium, saturated fat, protein, and fibre content of each purchase was provided by Kantar. Kantar also categorised product markets (e.g., breakfast cereals, chocolate) as high, mixed, or low in fruit, nuts, and vegetables, which we used to score products with 5 (>80%), 1 (>40% and $\leq$80%) or 0 ($\leq$40%) for fruit and vegetable content. The higher the final score, the less healthy the product. We applied the suggested cut-offs and considered food products that scored $\geq$4 points and drink

products that scored ≥1 point as HFSS [35]. Our primary outcomes were weekly household purchases of energy (kilocalories), fat (grams), saturated fat (grams), sugar (grams), and salt (grams) from HFSS products. We also examined the number of packs of HFSS products purchased. We were unable to assess volume purchased as this information was not available for all products.

## Statistical analysis

We used a CITS to estimate changes in the intervention group following the intervention compared to the counterfactual. We constructed the counterfactual by extrapolating the pre-intervention trend of the intervention group (based on 36 weeks) and incorporating the post-intervention changes of the control group (based on 44 weeks).

Our dataset contained a percentage of zero values for all outcomes because households did not purchase HFSS products every week. This percentage ranged from 2.8% for total HFSS products to 86.1% for sugary cereals. To account for this zero-inflation, we used a 2-part model for mixed discrete–continuous outcomes [38]. This type of model has been used previously to analyse positively skewed behavioural outcomes with a large proportion of zeros [39]. The 2-part model estimated the probability of purchasing a product (part 1—logit model) and, if a product was purchased, how much was purchased (part 2—generalised linear model). A gamma distribution was used for the analysis of energy and nutrients, whilst a negative binomial distribution was used for the analysis of packs. We used cluster-robust standard errors to account for clustering of outcomes by household in all models.

For each outcome, we used a single CITS model containing intervention and control data, with an indicator variable ('London', where intervention group = 1 and control group = 0). Our CITS models also included the following variables: time (time elapsed since the start of the study, expressed as week 1–80), a dummy variable ('intervention') indicating the pre-intervention period (coded 0) and the post-intervention period (coded 1), and interaction terms that accounted for the trend in the intervention group (time × London), the post-intervention period in the intervention group (intervention × London), the post-intervention trend in the control group (time × intervention), and the post-intervention trend in intervention group (time × intervention × London). All analyses were adjusted for household characteristics (number of adults and number of children) and sociodemographic characteristics of the main food shopper (sex, age, and socioeconomic position). We also included controls for season and festivals associated with HFSS purchasing (an indicator variable for weeks including Valentine's Day, Easter, Halloween, and Christmas).

From these 2-part models, we estimated mean weekly household energy purchased from HFSS products and used pairwise comparisons to test the difference in marginal means in the intervention group compared to the counterfactual in the post-intervention period. This outcome combined the change in both level and slope over the post-intervention period. The same comparison was used to examine differences in fat, saturated fat, sugar, and salt purchased through HFSS products, and packs of HFSS products purchased. Linear comparisons of parameters were used to estimate the percentage change in average marginal effects compared to the counterfactual. We also compared the changes in the first (25 February 2019 to 3 March 2019) and last (23 December 2019 to 29 December 2019) post-intervention weeks using linear comparisons of parameters, as an indication of the sustainability of any detected changes.

We used interaction terms to explore whether changes in energy purchased differed according to (1) socioeconomic position, (2) whether there were children in the household, and (3) whether the main food shopper was living with overweight or obesity (n = 1,591). However,

these analyses were limited by sample size, missing values in the case of BMI, and the uneven distribution of households within categories. Results from sub-group analyses are therefore descriptive and hypothesis generating.

Our results are reported relative to the counterfactual. We present marginalised results in the main paper. Coefficients from the underlying models are available in S2–S7 Tables. All analyses were conducted in Stata SE 16.

## Sensitivity analyses

**Analysis of a sub-sample of 'regular reporters' ($n$ = 1,126).** To test the influence of non-reporting on the results, we undertook analyses using regular reporters only. We defined regular reporters as households that reported purchases of any food or drink in more than 72 (90.0%) of the study weeks.

**Methodological triangulation.** Whilst the 2-part model enabled us to deal with zero-inflation, it does not fully account for the longitudinal panel nature of the data. To test the consistency of our findings across statistical models, we also fitted a mixed-effects negative binomial model. This model is appropriate for analysing skewed panel data but does not account for zero-inflation.

**Temporal falsification.** To test whether the observed changes were specific to the time the intervention occurred, we moved the 'intervention' week. If the changes were robust to the date of the intervention, we would expect to observe no changes at other times. We moved the 'intervention' from the week commencing 25 February 2019 to the week commencing 24 September 2018. This false intervention week was chosen because it was outside of any festival period (Valentine's Day, Easter, Christmas, and Halloween) and prior to the intervention.

**Changes in purchases by transport use.** We undertook an analysis of the sub-group of the main analytic sample ($n$ = 1,296) that reported their typical public transport use. This allowed us to examine whether regular TfL users (i.e., those typically using public transport at least once a week in London)—who likely had higher exposure to advertising on the TfL network—had greater changes in their HFSS purchases, thus increasing the likelihood that any observed changes were the result of the intervention.

**Restricting the time series.** We removed the last 2 weeks of data (16 to 29 December 2019), which represented a peak in HFSS purchases associated with Christmas, to see if this affected our overall findings.

**Changes in non-HFSS purchases.** We examined changes in mean weekly household purchases of non-HFSS products to see if there were any spillover effects on products not affected by the TfL policy.

## Ethics

Ethical approval for this study was granted by the London School of Hygiene & Tropical Medicine Ethics Committee (ref no: 16297/RR/11721). Written informed consent was obtained from all panel participants.

## Changes to protocol

Our original protocol specified a follow-up period of 12 months post-intervention. Follow-up was conservatively reduced to 10 months to avoid contamination of outcomes as a result of the early stages of the COVID-19 pandemic (reductions in the use of public transport and the early stages of panic buying). We conducted sensitivity analyses (described above) that were not pre-specified in our protocol, to assess the robustness of our results [40].

### Patient and public involvement

Patients and the public were not involved in the design, conduct, reporting, or dissemination plans of our research.

## Results

### Study population

Overall, 1,970 households were included in the analysis, with 977 households in London (intervention) and 993 households in the North of England (control) (Table 1). The intervention and control samples were similar in household characteristics but had small differences in main food shopper characteristics: sex (71.6% versus 74.3% female), socioeconomic position (27.5% versus 19.6% high socioeconomic position), and BMI (44.9% versus 53.1% overweight/obese). Intervention and control households also differed in transport use (67.8% versus 38.3% typically used public transport, of those who responded to the survey question). These differences are in line with differences seen between these regions [41–43]. We previously found that 45.5% of London participants and 28.8% of North of England participants reported seeing HFSS advertising on a transport network in the last 7 days, in a survey conducted in February 2019, prior to the introduction of the TfL advertising restrictions [8].

### Food and drink purchases

In total, 1,952,083 (38.4%) packs of food and drink purchased over the study period were classified as HFSS. Mean weekly household energy purchased from HFSS products was higher among control households (15,776.6 kcal pre-intervention and 15,697.3 kcal post-intervention) than among intervention households (14,199.7 kcal pre-intervention and 13,990.8 kcal post-intervention) overall, and for each HFSS category (Table 2). The CITS design assumes that the difference between the intervention group and control group is constant over time in

**Table 1. Descriptive characteristics of the overall, intervention, and control samples.**

| Characteristic | Sub-category | Total (N = 1,970) | Intervention (N = 977) | Control (N = 993) |
|---|---|---|---|---|
| **Household characteristics** | | | | |
| Number of adults in the household, mean (SD) | | 2.1 (0.9) | 2.1 (1.0) | 2.1 (0.8) |
| Number of children in the household, mean (SD) | | 0.5 (0.9) | 0.5 (0.9) | 0.5 (0.9) |
| Children in the household, n (%) | Yes | 575 (29.2) | 279 (28.6) | 296 (29.8) |
| | No | 1,395 (70.8) | 698 (71.4) | 697 (70.2) |
| **Main food shopper characteristics** | | | | |
| Sex, n (%) | Male | 533 (27.1) | 278 (28.5) | 255 (25.7) |
| | Female | 1,437 (72.9) | 699 (71.6) | 738 (74.3) |
| Age (years), mean (SD) | | 52.0 (13.8) | 52.1 (13.0) | 52.0 (14.6) |
| Socioeconomic position, n (%) | High | 464 (23.6) | 269 (27.5) | 195 (19.6) |
| | Middle | 1,164 (59.1) | 560 (57.3) | 604 (60.8) |
| | Low | 342 (17.4) | 148 (15.2) | 194 (19.5) |
| Body mass index, n (%) | Not overweight | 625 (31.7) | 337 (34.5) | 288 (29.0) |
| | Overweight/obese | 966 (49.0) | 439 (44.9) | 527 (53.1) |
| | Missing | 379 (19.2) | 201 (20.6) | 178 (17.9) |
| Public transport use, n (%) | No | 633 (32.1) | 182 (18.6) | 451 (45.4) |
| | Yes | 663 (33.7) | 383 (39.2) | 280 (28.2) |
| | Missing | 674 (34.2) | 412 (42.2) | 262 (26.4) |

**Table 2.  Unadjusted weekly household mean (SD) energy (kilocalories) purchased from high fat, salt, and sugar (HFSS) products and non-HFSS products pre- and post-intervention in the intervention group and control group.**

| Category | Pre-intervention weekly household mean | | | Post-intervention weekly household mean | | |
|---|---|---|---|---|---|---|
| | Total ($N = 1,970$) | Intervention ($N = 977$) | Control ($N = 993$) | Total ($N = 1,970$) | Intervention ($N = 977$) | Control ($N = 993$) |
| Total food & drink | 27,999.3 (20,661.4) | 26,818.5 (21,500.1) | 29,133.3 (19.756.6) | 27,763.5 (20,215.0) | 26,507.7 (20,872.2) | 28,989.7 (19.474.4) |
| Total HFSS products | 15,004.1 (13,234.2) | 14,199.7 (13,694.7) | 15,776.6 (12,728.8) | 14,854.3 (12,959.5) | 13,990.8 (13,228.2) | 15,697.3 (12,635.1) |
| Chocolate & confectionery | 1,445.4 (2,549.5) | 1,266.9 (2,439.6) | 1,616.8 (2,639.4) | 1,504.5 (2,542.7) | 1,308.9 (2,406.3) | 1,695.5 (2,655.4) |
| Puddings & biscuits | 3,071.1 (3,892.7) | 2,827.2 (3,754.0) | 3,305.3 (4,007.6) | 3,071.5 (3,814.6) | 2,795.9 (3,693.4) | 3,340.6 (3,910.7) |
| Sugary drinks | 248.2 (682.1) | 232.0 (668.0) | 263.9 (695.1) | 221.7 (637.3) | 211.2 (625.3) | 232.0 (648.7) |
| Sugary cereals | 467.8 (1,426.5) | 461.7 (1,496.0) | 473.7 (1,356.4) | 418.8 (1,323.1) | 405.6 (1,346.9) | 431.7 (1,299.4) |
| Savoury snacks | 1,075.0 (1,693.0) | 1,046.7 (1,698.8) | 1,102.1 (1,686.9) | 1,085.3 (1,702.0) | 1,073.8 (1,732.2) | 1,096.4 (1,672.0) |
| Non-HFSS products | 12,995.2 (9,860.4) | 12,618.9 (10,442.5) | 13,356.6 (9,252.8) | 12,909.3 (9,672.6) | 12,516.9 (10,195.1) | 13,292.4 (9,117.6) |

the absence of the intervention [44]. We tested this assumption using the significance of the London × time interaction term in our models. The pre-intervention trend in purchasing followed a similar pattern in both regions for all HFSS categories, except chocolate and confectionery (S1 Fig; S8 Table). For chocolate and confectionery, there was a parallel trend in the later part of the pre-intervention period (from 19 November 2018), but some deviation in the earlier weeks (S9 Table).

## Changes in total HFSS purchases

The implementation of HFSS advertising restrictions was associated with a relative reduction in average weekly household energy purchased from HFSS products of 1,001.0 kcal (95% CI 456.0 to 1,546.0), or 6.7% (95% CI 3.2% to 10.1%), in the intervention group compared to the counterfactual (Fig 3; Table 3). Relative decreases in weekly household purchases of fat (57.9 g; 95% CI 22.1 to 93.7), saturated fat (26.4 g; 95% CI 12.4 to 40.4), and sugar (80.7 g; 95% CI 41.4 to 120.1) from HFSS products, and packs of HFSS products purchased (0.7 packs; 95% CI 0.2 to 1.2), were also observed. The observed relative change in weekly household purchases of salt was not significant (−2.2 g, 95% CI −9.8 to 5.4). The relative reduction in energy purchased detected in the final post-intervention week was larger than that detected in the first week post-intervention (see S10 Table).

## Changes by HFSS category

**Chocolate and confectionery.**   Using the whole study period, average weekly household purchases of energy from chocolate and confectionery fell by 317.9 kcal (95% CI 200.0 to 435.8) in the intervention group relative to the counterfactual, equivalent to an observed decrease of 19.4% (95% CI 13.4% to 25.4%). Relative reductions in weekly household purchases of fat (13.1 g; 95% CI 7.5 to 18.8), saturated fat (8.7 g; 95% CI 5.7 to 11.7), sugar (41.4 g; 95% CI 27.4 to 55.4), and salt (0.2 g; 95% CI 0.1 to 0.2) were also detected in the post-intervention period. A relative reduction in the number of packs of HFSS products purchased (0.4; 95% CI 0.3 to 0.6) was also observed. We re-ran this model using a shorter pre-intervention period to satisfy the parallel trends assumption and observed similar changes (S11 Table).

**Puddings and biscuits.**   Relative to the counterfactual, energy purchased from puddings and biscuits was lower in the intervention group in the post-intervention period (198.3 kcal; 95% CI 6.9 to 389.7). A relative reduction in purchased salt was also observed (0.4 g; 95% CI 0.1 to 0.8). However, relative changes in the amount of fat (−8.1 g; 95% CI −16.5 to 0.4), saturated fat (−2.9 g; 95% CI −7.4 to 1.6), and sugar (−7.8; 95% CI −23.7 to 8.2) purchased through

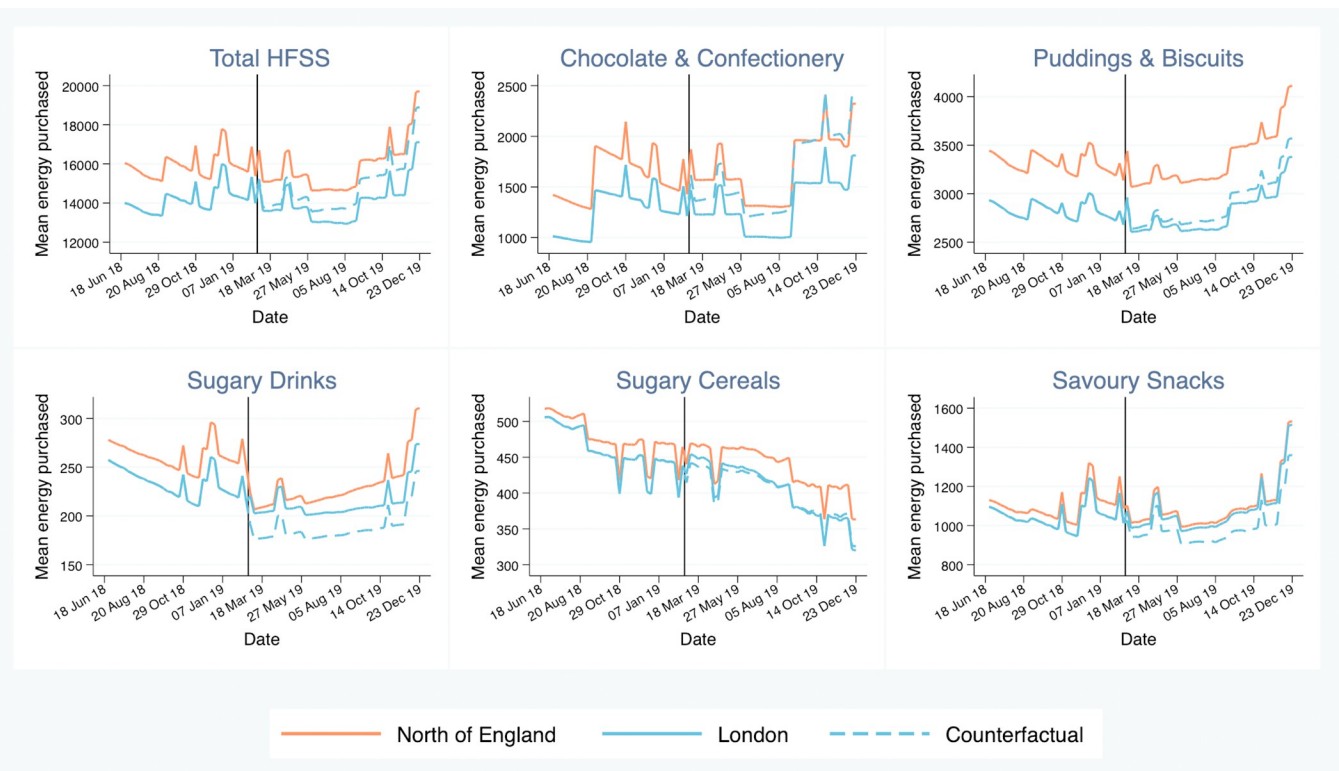

**Fig 3. Adjusted weekly household mean energy purchased from high fat, salt, and sugar (HFSS) products in London (intervention), the North of England (control), and the counterfactual.** Vertical line = date of intervention implementation. The counterfactual was estimated by extrapolating the pre-intervention trend in London and incorporating the post-intervention changes in the North of England. Weekly household mean energy purchased from HFSS products was estimated from a controlled interrupted time series 2-part model: part 1 (logit) and part 2 (generalised linear model) with gamma distribution. Models were adjusted for festivals, season, number of adults in household, number of children in household, and sex, age, and socioeconomic position of main food shopper. Cluster-robust standard errors were used. Household-week observations where households did not report any food and drink purchases that week were dropped. Data period = 18 June 2018 to 29 December 2019. Spikes represent festival weeks included in the models.

puddings and biscuits, and the number of packs of puddings and biscuits purchased (−0.1 packs, 95% CI −0.3 to 0.1), were non-significant.

**Sugary drinks and sugary cereals.** Small, non-significant relative increases were observed for energy purchased from sugary drinks (19.6 kcal; 95% CI −14.0 to 53.3) and sugary cereals (4.1 kcal; 95% CI −88.7 to 96.8) following the intervention. There were also no significant differences detected for nutrients and packs purchased.

**Savoury snacks.** There was a non-significant relative increase in energy purchased from savoury snacks (61.1 kcal; 95% CI −20.3 to 142.5) following the intervention. The average weekly number of packs purchased increased marginally (0.1 packs; 95% CI 0.0 to 0.2) relative to the counterfactual. This was accompanied by a small relative increase in salt purchased through savoury snacks (0.4 g; 95% CI 0.1 to 0.7). Relative increases for other nutrients were not significant.

## Changes by household and sociodemographic characteristics

There was some indication that observed differences varied by population sub-group, but these did not reach statistical significance. Descriptively, for example, we observed the largest relative reduction in purchased energy in the middle socioeconomic group (S12 Table). The relative reduction was larger in households with children for chocolate and confectionery

**Table 3. Changes and percentage changes in weekly household mean (95% CI) energy and nutrients purchased from high fat, salt, and sugar (HFSS) products and packs of HFSS products purchased, in London (intervention group) compared to the counterfactual, 18 June 2018 to 29 December 2019 (*n* = 1,970).**

| Outcome | Measure | Total HFSS products | Chocolate & confectionery | Puddings & biscuits | Sugary drinks | Sugary cereals | Savoury snacks |
|---|---|---|---|---|---|---|---|
| Energy | Kilocalories | **−1,001.0 (−1,546.0 to −456.0)** | **−317.9 (−435.8 to −200.0)** | **−198.3 (−389.7 to −6.9)** | 19.6 (−14.0 to 53.3) | 4.1 (−88.7 to 96.8) | 61.1 (−20.3 to 142.5) |
| | Percent | **−6.7 (−10.1 to −3.2)** | **−19.4 (−25.4 to −13.4)** | **−6.6 (−12.6 to −0.6)** | 10.1 (−8.8 to 29.1) | 1.0 (−22.5 to 24.5) | 6.0 (−2.4 to 14.5) |
| Fat | Grams | **−57.9 (−93.7 to −22.1)** | **−13.1 (−18.8 to −7.5)** | −8.1 (−16.5 to 0.4) | −0.2 (−1.0 to 0.7) | −0.5 (−3.3 to 2.3) | 3.9 (−0.9 to 8.7) |
| | Percent | **−6.5 (−10.4 to −2.7)** | **−18.2 (−24.9 to −11.6)** | **−6.2 (−12.4 to −0.1)** | −6.4 (−36.4 to 23.6) | 5.8 (−35.5 to 23.9) | 6.8 (−2.2 to 15.7) |
| Saturated fat | Grams | **−26.4 (−40.4 to −12.4)** | **−8.7 (−11.7 to −5.7)** | −2.9 (−7.4 to 1.6) | −0.1 (−0.8 to 0.6) | −0.3 (−1.1 to 0.5) | 0.8 (−0.1 to 1.7) |
| | Percent | **−7.3 (−11.0 to −3.7)** | **−22.8 (−29.2 to −16.4)** | −4.5 (−11.2 to 2.2) | −5.4 (−38.4 to 27.6) | −10.9 (−38.5 to 16.8) | 9.2 (−2.0 to 20.4) |
| Sugar | Grams | **−80.7 (−120.1 to −41.4)** | **−41.4 (−55.4 to −27.4)** | −7.8 (−23.7 to 8.2) | 4.9 (−0.9 to 10.7) | 0.7 (−5.0 to 6.5) | 0.8 (−0.4 to 1.9) |
| | Percent | **−10.5 (−15.2 to −5.9)** | **−21.8 (−27.8 to −15.7)** | −3.3 (−10.0 to 3.3) | 13.9 (−4.6 to 32.4) | 3.0 (−21.7 to 27.7) | **8.4 (5.0 to 21.8)** |
| Salt | Grams | −2.2 (−9.8 to 5.4) | **−0.2 (−0.2 to −0.1)** | **−0.4 (−0.8 to −0.1)** | −0.0 (−0.1 to 0.1) | 0.0 (−0.1 to 0.2) | **0.4 (0.1 to 0.7)** |
| | Percent | −3.6 (−15.5 to 8.3) | **−17.4 (−25.5 to −9.2)** | **−12.0 (−19.6 to −4.4)** | −11.6 (−41.5 to 18.4) | 9.0 (−17.1 to 35.2) | **12.5 (2.4 to 22.6)** |
| Packs | Number | **−0.7 (−1.2 to −0.2)** | **−0.4 (−0.6 to −0.3)** | −0.1 (−0.3 to 0.1) | 0.0 (−0.1 to 0.1) | −0.0 (−0.1 to 0.0) | **0.1 (0.0 to 0.2)** |
| | Percent | **−4.9 (−8.4 to −1.4)** | **−21.4 (−28.0 to −14.8)** | −1.9 (−8.7 to 4.8) | 4.4 (−11.2 to 20.1) | −7.4 (−27.9 to 13.2) | 9.0 (−0.0 to 17.9) |

Bold indicates significant at the 95% confidence level. Weekly household mean purchases were estimated from a controlled interrupted time series 2-part model: part 1 (logit) and part 2 (generalised linear model), with gamma distribution for energy and nutrients and negative binomial distribution for packs. Models were adjusted for festivals, season, number of adults in household, number of children in household, and sex, age, and socioeconomic position of main food shopper. Cluster-robust standard errors were used. Household-week observations where households did not report any food and drink purchases that week were dropped. Data period = 18 June 2018 to 29 December 2019.

purchases, but larger in households with no children for total HFSS and puddings and biscuits (S13 Table). Reductions were larger for households with a main food shopper who was living with overweight or obesity for total HFSS, chocolate and confectionery, and puddings and biscuits (S14 Table). However, these results should be interpreted with caution and can be considered hypothesis generating. Studies with greater power are required to explore these associations further.

## Sensitivity analyses

Results for regular reporters were comparable to those for the full sample (S15 Table). Similar or larger changes were also observed using mixed-effects negative binomial models compared to the analyses presented above (S16 Table). When the date of intervention implementation was changed, no changes in purchasing were detected, providing strong evidence that the observed changes are robust to the time of implementation (S17 Table). There was some indication that changes were greater among households where the main food shopper reported public transport use, though these results were non-significant (S18 Table). When the last 2 weeks of the study period were removed, the observed changes remained similar (S19 Table). These sensitivity analyses provide additional support for the robustness of our results. No significant changes in non-HFSS purchases were detected in intervention households relative to the counterfactual in the post-intervention period (S20 Table), suggesting no spillover effect on non-HFSS purchases.

## Discussion

Compared to the counterfactual, this study found that the introduction of advertising restrictions for HFSS products across the London transport network was associated with a relative decrease in average weekly household purchases of energy from HFSS products of 6.7%, or 1,001.0 kcal. Using the mean household size of 2.6 people in the sample, and assuming an even energy distribution, this equates to a relative reduction in purchased energy of 385.0 kcal per

person per week, which is equivalent to approximately 72.1 g of standard milk chocolate. Relative reductions in weekly household purchases of fat (57.9 g), saturated fat (26.4 g), and sugar (80.7 g) from HFSS products were also observed. The magnitude of the observed change in sugar purchased associated with the TfL policy is larger than that reported for the UK Soft Drinks Industry Levy (SDIL), a population-level policy widely regarded as successful, which reduced weekly household purchases of sugar by 29.5 g [45]. We observed the largest relative reduction for chocolate and confectionery (19.4%; 317.9 kcal). However, decreases associated with the intervention are in the context of underlying increases in purchases of HFSS products in both the intervention and control areas over the study period. This means that the intervention was associated with a reduced rate in growth of HFSS purchases in the intervention group rather than an absolute reduction in HFSS purchases.

## Strengths and limitations

We used a CITS study design as a robust approach to evaluate a natural experiment where a randomised design was not feasible or pragmatic [35]. Such studies are conducted in real-world settings [35,40], and can provide evidence to inform policy [46]. Use of a control group reduced the risk of national-level, time-varying confounders driving observed results, such as seasonal effects, underlying trends in HFSS purchasing, and the effect of other sugar and calorie reduction strategies such as the SDIL [47–49]. Confounding due to other events occurring at the same time as the intervention in either the intervention or control group cannot be ruled out. The changes detected also do not pass the Bonferroni threshold (i.e., $P \sim 0.001$, based on $P = 0.05/36$ tests). However, our sensitivity analyses point to our findings being specific to the time of intervention implementation and only detected for HFSS products, especially for the HFSS category chocolate and confectionery, which has few non-HFSS substitutes. There was also an indication that relative reductions observed were larger among regular public transport users, who likely had a larger change in advertising exposure as a result of the TfL policy. Our sensitivity analyses therefore provide support for the observed changes being associated with the TfL policy rather than other events occurring at the same time, or occurring by chance.

The parallel trends assumption was met for all outcomes except chocolate and confectionery. However, similar magnitudes of change were observed when restricting the chocolate and confectionery sample to a shorter pre-intervention time period that exhibited parallel trends. This suggests that the changes observed for chocolate and confectionery were not affected by the lack of parallel trends at the earlier time points. We also found no changes in weekly household purchases of non-HFSS products associated with the introduction of the policy, suggesting that substitution of non-HFSS product categories was unlikely.

Included households were representative of the regions studied in terms of sex, age, socioeconomic position, and household size. Purchase data have been found to be an accurate estimate of food consumption [50]. Most households did not report purchases every week, and we assumed non-reporting was random. However, missingness may have been associated with purchasing behaviour (e.g., forgetting to report purchases from smaller shopping occasions or choosing not to report purchases from less healthy shopping occasions). This study only considered take-home grocery purchases. Out-of-home purchase data were available for a subset of the Kantar FMCG panel, but we did not include these due to limited nutritional data available for out-of-home purchases and differences in data collection methodology.

Companies may have designed advertisements to be compliant with the TfL regulations and then used these in other locations and media, resulting in contamination in the control area. This would result in underestimation of the effect. As the policy was first subject to a

public consultation in May 2018, then announced in November 2018, ahead of implementation in February 2019, companies may have also adapted their advertising before the implementation date, but this is unlikely due to the lead-in time for campaigns and the duration of existing campaigns. Our sensitivity analyses suggest observed decreases are robust to time of implementation.

## Comparison with other studies

There are few directly comparable studies on the impact of advertising restrictions on purchasing [20,21,26,28,51]. One international study compared broadcast advertising policies across countries, finding that countries with statutory restrictions had reduced volume of 'junk food' sales per capita, whilst countries with no policy or voluntary policies had increased volumes of 'junk food' sales per capita [19]. A study conducted in the UK comparing periods of time with no television advertising restrictions (pre-2004), industry self-regulation (2005–2007), and statutory regulation (post-2007) found that statutory regulations were effective in reducing household purchases of HFSS products (by £5.60 for drinks and £14.90 for food per capita per quarter among households with children) [52]. Evaluations of restrictions on unhealthy food advertising to children in Quebec and Singapore found decreases in the likelihood of consuming fast food [27,53]. An evaluation of the Chilean food marketing policy found a decrease in household purchases of sugary drinks [51]. However, there was no evidence to suggest that a reduction in advertising exposure mediated a reduction in HFSS consumption, which may have been caused by other factors related to the policy [26].

## Interpretation and policy implications

The observed relative reductions in energy, fat, and sugar purchased from HFSS products associated with the restriction of HFSS product advertising across the TfL network could have a meaningful impact on population health. A recent modelling study in the UK estimated that restricting HFSS advertising on television from 5:30 AM to 9:00 PM has the potential to reduce daily energy intake by 9.1 kcal for UK children and result in 40,000 fewer children with obesity [54]. Another modelling study predicted a 24.6 kcal reduction in daily household energy purchased if the price of sugary snacks increased by 20% [55]. The reductions observed in our study were larger than both of these previous estimates, equating to around 55.0 kcal per person (or 143.0 kcal per household) per day. The reduction of 80.7 g of sugar per household per week estimated in our study is also larger than that reported for the SDIL (29.5 g per household per week) [45]. This suggests that the TfL policy has the potential to be a highly effective intervention.

Advertisements were vetted by TfL and only approved if compliant with the policy. For brands selling products with no policy-compliant alternatives (e.g., chocolate and confectionery), advertising was no longer possible. The intervention 'dose' was therefore plausibly strongest for chocolate and confectionery, where we found the largest change in purchases. An exceptions process allowed companies to apply to advertise a HFSS product by providing evidence that the product does not contribute to childhood obesity. This mechanism has not been widely used, which suggests that companies are not directly challenging the policy. However, other tactics to circumvent the policy could have been used. For brands with policy-compliant alternative products (e.g., low or zero calorie drinks), brand advertising continued even though advertisement of specific products was restricted. This may explain the lack of change in sugary drink, sugary cereal, and savoury snack purchases—product categories that often have non-HFSS substitutes. Brands ceasing advertising because they had no policy-compliant products may have also created advertising space for brands that have policy-compliant

products, allowing them to increase visibility and penetration of their brands [56]. Brand advertising has been found to elicit a brain response and to increase consumption of unhealthy foods and drinks, even when the advertised product is healthy [57,58]. This indicates the potential importance of restricting brand advertising, in addition to individual product advertising, in order to optimise the effectiveness of the policy. It may also be beneficial to restrict advertising of certain nutrient-poor product categories (e.g., cakes and energy bars) regardless of the individual product's NPM score [59].

We observed larger relative reductions in the last week of the post-intervention period compared to the first. This could be because of delays in the removal of existing HFSS advertising. Consumer behaviour change may also require a longer time frame to shift because food preferences can be difficult to change and there are strong associations between certain brands and HFSS products [60,61]. Although we were unable to assess the sustainability of the observed relative reductions beyond 44 weeks, the larger changes over time indicate that the changes could plausibly be sustained, and may even increase, over time. However, further studies are needed to confirm this.

Sub-group descriptive analyses provide some indication that relative reductions in purchases were greater in less affluent households and households where the main food shopper was living with overweight or obesity. However, these were not statistically significant. If these findings are confirmed in other studies, the policy would be well-targeted to households that would benefit the most from this intervention, and may help reduce inequalities in diet [5,8,9]. This supports previous work that suggests population-level policies are more effective and equitable [62,63]. However, further confirmatory research is required.

The policy was associated with attenuated growth of HFSS purchases rather than an absolute reduction in HFSS purchases. Single interventions cannot be expected to work on their own and should be seen as one part of a wider strategy to improve population health, with multiple interventions needed at multiple points within the food system to improve diet [64,65]. In the UK, the recently proposed restrictions on HFSS advertising before the television '9:00 PM watershed' and on all online HFSS advertising [66], coupled with restrictions on outdoor HFSS advertising being considered in other locations [67], may be an example of emerging policy coherence in this area. This will likely create an overall healthier advertising environment by limiting the displacement of HFSS advertising across advertising media [18,24].

### Future research

Future studies need to explore the possible longer-term effects of HFSS advertising restriction policies. Monitoring the response of brands and advertisers and their adaptiveness to a changed policy environment is important in order to optimise and design future policy. For example, the policy may stimulate further product reformulation or the use of unregulated advertising media, or prompt companies to focus more on brand advertising [56,68]. Future studies should also explore the impact of advertising restrictions on out-of-home purchases of HFSS products, which may reveal even further reductions. Studies to quantify the potential impact of such interventions on obesity and related diseases are needed. Replication of this study elsewhere is also important as apparent effect sizes may be lower in settings outside of London where routine use of public transport is lower and the outdoor advertising estate is smaller [69]. In addition, better powered studies to assess differential effects on population sub-groups and inequalities are required. Studies that explore the mechanisms behind changes in food and drink purchases associated with advertising policies, such as changes in advertising exposure, are also important.

## Conclusions

In the 10 months following the introduction of the TfL's HFSS advertising restrictions, we observed a relative reduction in average weekly household purchases of energy from HFSS products of 6.7%, or 1,001.0 kcal. This included a 19.4% (317.9 kcal) reduction for chocolate and confectionery. These findings provide support for policies that restrict HFSS product advertising as a tool to reduce purchases of HFSS products, as a way of improving population diet and preventing obesity.

## Supporting information

**S1 Fig. Unadjusted weekly household mean energy purchased from HFSS products over the study period.** Vertical line = date of intervention implementation.
(TIF)

**S1 Table. Definition of HFSS categories.**
(DOCX)

**S2 Table. Coefficients for 2-part model: Energy.**
(DOCX)

**S3 Table. Coefficients for 2-part model: Fat.**
(DOCX)

**S4 Table. Coefficients for 2-part model: Saturated fat.**
(DOCX)

**S5 Table. Coefficients for 2-part model: Sugar.**
(DOCX)

**S6 Table. Coefficients for 2-part model: Salt.**
(DOCX)

**S7 Table. Coefficients for 2-part model: Packs.**
(DOCX)

**S8 Table. Difference in pre-intervention trends in energy (kilocalories) purchased from HFSS products in London (intervention) and the North of England (control).**
(DOCX)

**S9 Table. Difference in pre-intervention trend for energy (kilocalories) purchased from chocolate and confectionery in London (intervention) and the North of England (control) using different start dates.**
(DOCX)

**S10 Table. Changes in weekly household mean (95% CI) energy (kilocalories) purchased from HFSS products, and packs of HFSS products purchased, in London (intervention group) compared to the counterfactual in the first and last post-intervention week ($n$ = 1,970),**
(DOCX)

**S11 Table. Changes in weekly household mean (95% CI) energy and nutrients purchased from chocolate and confectionery, and packs of chocolate and confectionery purchased, in London (intervention group) compared to the counterfactual using 2 pre-intervention periods.**
(DOCX)

**S12 Table. Changes in weekly household mean (95% CI) energy (kilocalories) purchased from HFSS products, in London (intervention group) compared to the counterfactual, among high socioeconomic households, and additional changes among middle and low socioeconomic households (*n* = 1,970).**
(DOCX)

**S13 Table. Changes in weekly household mean (95% CI) energy (kilocalories) purchased from HFSS products, in London (intervention group) compared to the counterfactual, among households with no children, and additional changes among households with children (*n* = 1,970).**
(DOCX)

**S14 Table. Changes in weekly household mean (95% CI) energy (kilocalories) purchased from HFSS products, in London (intervention group) compared to the counterfactual, among households with a main food shopper not living with overweight or obesity, and additional changes among households with a main food shopper living with overweight or obesity (*n* = 1,591).**
(DOCX)

**S15 Table. Changes in weekly household mean (95% CI) energy and nutrients purchased from HFSS products, and packs of HFSS products purchased, in London (intervention group) compared to the counterfactual using a sub-sample of regular reporters (*n* = 1,126).**
(DOCX)

**S16 Table. Changes in weekly household mean (95% CI) energy and nutrients purchased from HFSS products, and packs of HFSS products purchased, in London (intervention group) compared to the counterfactual estimated using a mixed-effects negative binomial model (*n* = 1,970).**
(DOCX)

**S17 Table. Temporal falsification sensitivity analysis with intervention week moved to the week commencing 24 September 2018 (*n* = 1,970).**
(DOCX)

**S18 Table. Changes in weekly household mean (95% CI) energy (kilocalories) purchased from HFSS products, in London (intervention group) compared to the counterfactual, among regular public transport users, and additional changes among those who are not regular users (*n* = 1,296).**
(DOCX)

**S19 Table. Changes and percentage changes in weekly household mean (95% CI) energy and nutrients purchased from HFSS products, and packs of HFSS products purchased, in London (intervention group) compared to the counterfactual. Data period = 18 June 2018 to 15 December 2019.**
(DOCX)

**S20 Table. Changes in weekly household mean (95% CI) energy and nutrients purchased from non-HFSS products and packs of non-HFSS products purchased, in London (intervention group) compared to the counterfactual. Data period = 18 June 2018 to 29 December 2019.**
(DOCX)

**S1 Text. Study protocol.**
(DOCX)

**S2 Text. STROBE checklist.**
(DOCX)

## Acknowledgments

We thank Dr. Samantha Hajna for help with producing Fig 1.

## Author Contributions

**Conceptualization:** Amy Yau, Nicolas Berger, Cherry Law, Laura Cornelsen, Robert Greener, Jean Adams, Emma J. Boyland, Thomas Burgoine, Frank de Vocht, Matt Egan, Vanessa Er, Amelia A. Lake, Karen Lock, Oliver Mytton, Mark Petticrew, Claire Thompson, Martin White, Steven Cummins.

**Data curation:** Amy Yau.

**Formal analysis:** Amy Yau, Nicolas Berger, Cherry Law, Laura Cornelsen, Robert Greener, Steven Cummins.

**Funding acquisition:** Laura Cornelsen, Jean Adams, Emma J. Boyland, Thomas Burgoine, Frank de Vocht, Matt Egan, Amelia A. Lake, Karen Lock, Oliver Mytton, Mark Petticrew, Claire Thompson, Martin White, Steven Cummins.

**Investigation:** Amy Yau, Nicolas Berger, Cherry Law, Laura Cornelsen, Robert Greener, Jean Adams, Emma J. Boyland, Thomas Burgoine, Frank de Vocht, Matt Egan, Vanessa Er, Amelia A. Lake, Karen Lock, Oliver Mytton, Mark Petticrew, Claire Thompson, Martin White, Steven Cummins.

**Methodology:** Amy Yau, Nicolas Berger, Cherry Law, Laura Cornelsen, Robert Greener, Jean Adams, Emma J. Boyland, Thomas Burgoine, Frank de Vocht, Matt Egan, Vanessa Er, Amelia A. Lake, Karen Lock, Oliver Mytton, Mark Petticrew, Claire Thompson, Martin White, Steven Cummins.

**Visualization:** Amy Yau, Cherry Law.

**Writing – original draft:** Amy Yau, Steven Cummins.

**Writing – review & editing:** Nicolas Berger, Cherry Law, Laura Cornelsen, Robert Greener, Jean Adams, Emma J. Boyland, Thomas Burgoine, Frank de Vocht, Matt Egan, Vanessa Er, Amelia A. Lake, Karen Lock, Oliver Mytton, Mark Petticrew, Claire Thompson, Martin White.

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
