## [Editor Report · Decision Letter 0]

4 Oct 2021

Dear Dr Yau, 

Thank you for submitting your manuscript entitled "Changes in household food and drink purchases following restrictions on the advertisement of high fat, salt and sugar products across the Transport for London network: A controlled interrupted time series analysis" for consideration by PLOS Medicine.

Your manuscript has now been evaluated by the PLOS Medicine editorial staff and I am writing to let you know that we would like to send your submission out for external assessment.

However, we first need you to complete your submission by providing the metadata that is required for full assessment. To this end, please login to Editorial Manager where you will find the paper in the 'Submissions Needing Revisions' folder on your homepage. Please click 'Revise Submission' from the Action Links and complete all additional questions in the submission questionnaire.

Please re-submit your manuscript within two working days, i.e. by Oct 06 2021 11:59PM.

Once your full submission is complete, your paper will undergo a series of checks in preparation for assessment. 

Kind regards,

Richard Turner, PhD

rturner@plos.org

---

## [Decision Letter · Decision Letter 1]

27 Oct 2021

Dear Dr. Yau,

Thank you very much for submitting your manuscript "Changes in household food and drink purchases following restrictions on the advertisement of high fat, salt and sugar products across the Transport for London network: A controlled interrupted time series analysis" (PMEDICINE-D-21-04172R1) for consideration at PLOS Medicine. 

Your paper was discussed with an academic editor with relevant expertise and sent to independent reviewers, including a statistical reviewer. The reviews are appended at the bottom of this email and any accompanying reviewer attachments can be seen via the link below:

[LINK]

In light of these reviews, we will not be able to accept the manuscript for publication in the journal in its current form, but we would like to invite you to submit a revised version that addresses the reviewers' and editors' comments fully. You will appreciate that we cannot make a decision about publication until we have seen the revised manuscript and your response, and we expect to seek re-review by one or more of the reviewers. 

We hope to receive your revised manuscript by Nov 17 2021 11:59PM. Please email us (plosmedicine@plos.org) if you have any questions or concerns.

Please let me know if you have any questions, and we look forward to receiving your revised manuscript. 

Sincerely,

Richard Turner, PhD

Senior editor, PLOS Medicine

rturner@plos.org

Please restructure the abstract so that the final sentence of the "Methods and findings" subsection begins "Study limitations include ..." or similar and quotes 2-3 of the study's main limitations. 

Please add a sentence to the abstract around line 44 to quote the baseline differences in the samples (information around line 274 in the Results section). 

Please avoid claims such as "the first" at line 57, and where needed add "to our knowledge" or similar. 

After the abstract, please add a new and accessible "Author summary" section in non-identical prose. You may find it helpful to consult one or two recent research papers in PLOS Medicine to get a sense of the preferred style. 

Noting the study registration, is a protocol or prespecified analysis plan available? If so, please attach this as a supplementary file, referred to in the Methods section. 

Please highlight non-prespecified analyses. 

Please avoid language implying causality, throughout the text. For example, at line 54 in the abstract "observed effects" should be amended to "observed changes" or similar. 

Similarly, please adapt the wording to "Relative reductions ..." or similar at line 49, for example. 

Throughout the text, please locate reference call-outs prior to punctuation, e.g., "... HFSS products [1,2,4].".

Please remove the information on data sharing and funding from the end of the main text. In the event of publication, this information will appear in the article metadata, via entries in the submission form. 

In the reference list, please use the journal name abbreviations "PLoS Med." and "PLoS ONE".

For reference 3 and the other relevant citations, please remove the name of the paper's academic editor. 

Please include a completed STROBE checklist with your revision, labelled "S1_STROBE_Checklist" or similar and referred to as such in the Methods section (main text). 

In the checklist, please refer to individual items by section (e.g., "Methods") and paragraph number, not by line or page numbers as these generally change in the event of publication. 

Comments from the reviewers:

*** Reviewer #1: 

Thanks for the opportunity to review your manuscript. My role is as a statistical reviewer so my review focuses on the study design, data, and the analysis that is presented. I have put general queries first and followed those with questions relevant for a specific section of the manuscript.

This study uses data from a commercial panel study, and includes households in London and the north of England. The event of interest was the introduction of new advertising policy on the public transport system in London (TfL). The time series included 36 weeks of data before the event change in advertising, and 44 weeks after the event. A hurdle model (logit - bought item, GLM - how much purchased). The strengths of the study are the control region, a regression model well-suited to an 'energy purchased' variable, and measurements very specific to what the policy change aimed to achieve. The manuscript is well-written and has well-presented tables and figures and thoughtful sensitivity analyses.

There are some details in the pre-registered protocol, as a more detailed statistical analysis plan developed for the controlled ITS? The protocol also details a qualitative and impact evaluation - are these currently under review elsewhere? 

P2, L42. Could you clarify what each n refers to here? I.e. is the first n the number of records? The number of participants might be more useful to include.

P2, L46. I would drop the 'n=' from the study periods here and on line 47

P3, L91. Could you briefly describe what the 'TfL' network composes? I assume this includes advertising at train stations, bus shelters, and advertising on the public transport vehicles as well. 

P5, L121. Which areas in the 'North of England' were sampled? Can the postcodes or areal identifiers be included as an appendix? 

P5, L124. Perhaps worth being specific and describing it as a 'location-based control'?

P5, L125. So some households did not contribute data to the entire time-series? How many households had complete data across the study period? 

P5, L127. 'global seasonal fluctuation' might be a better description as it's impossible to exclude a seasonal effect unique to one of the two study locations. 

P6, L191. Is this the Huber 'sandwich' robust standard error? 

P8, L221. Just to check - these are from the 'margins' command in Stata? For my own interest - which Stata command did you use to estimate the two-part model that was compatible with margins?

Fig 2. Fascinating figures - it turns out my vision of the English tucking into pudding and chocolates over the chilly Christmas period is completely correct. The line for the counterfactual seems to get lost in this panel (e.g. sugary cereals/chocolate). I can't give you a specific recommendation to improve this but perhaps it's worth taking a second look at these figures?

Supp Table 12. Where these estimated by fitting a separate model for each sub-group? One limitation of this approach is that you shrink the sample size and so whether specific subgroup has a 'significant' treatment effect then depends strongly on how much sample size there is in it (and relies on 'differences in nominal significance'). An alternative is to include the sub-group variable in the regression model and look at the interaction of this with the relevant parameters to test for overall heterogeneity. I have to admit I am not sure how or if this could be achieved easily with the two-stage model you are using. You are fairly careful about describing these analyses but I would consider being clear about the limitation of a stratified analysis rather than one with an interaction to look for sub-group heterogeneity. 

*** Reviewer #2: 

This study evaluated the implementation of an intervention to restrict advertisement of HFSS products in public transport using household purchasing data. The paper is well written although there are several points which need clarification. A major concern is the comparability of the intervention and control sample, especially given the absolute differences in baseline purchasing patterns between the samples. 

Specific comments:

Introduction

Line 84-88 Can the authors describe what other locations they mean here? In general there's enough background on the HFSS on TV but no much on public transport or other relevant environments

Methods

Line 125-128 How comparable is the control sample of households in terms of exposure to advertisement through public transportation? A bit more description of this is needed. As seen in Box1, TfL is a very large network which means people may be highly exposed to advertisements whereas in the control area this may not be the case, perhaps limiting the comparability of the chosen sample? 

Are there any other criteria for matching (if this was done) intervention and control households?

Line 140 Can the mean study weeks include the IQR (instead/in addition to SD)

Line 144 Can the % missing be reported here

Line 151-152 Can the authors estimate the % imputed values in this study - instead of referencing other studies?

Line 166 Why is the variable for public transport use (yes/no) used in a sensitivity analysis, rather than a secondary or exploratory outcome? Was this not part of the initial protocol?

Line 174 The description of the main nutrient outcomes could be clearer. Is this kcal from HFSS/household/week?

Have the author considered using mean food expenditure/week as an additional outcome?

Statistical analysis. The choice of models and sensitivity analyses seem robust. However, I wonder why the authors haven't included an analysis using monthly purchases instead of weekly. I think that there could be a period of time (perhaps several weeks) from intervention implementation at which effects are detectable. Perhaps for something like advertisement, removal of HFSS ads may not have an immediate effect on purchases and a longer period of time may need to be considered.

Results

Table 2 doesn't seem to be sufficiently described in the manuscript. There are absolute differences before policy implementation between intervention and control samples, especially for chocolate & conf. Where these baseline purchasing patterns tested? Large differences between intervention and control samples may limit their comparability. 

Results/discussion

In general, the authors talk about the "reduction" in HFSS observed after the intervention implementation, whereas looking at the trends, it feels like it's rather an attenuation of the growth that would be expected in normal circumstances. I would advise to revise the text accordingly

Is there an explanation for the results observed in sugary drinks/cereals and savoury snacks?

*** Reviewer #3: 

General Comments: The study evaluates whether purchases of typically HFSS products improved in nutritional measures before vs after the TfL's advertising policy was implemented in Feb 2019 focusing on Londoners' (and also Londoners' who reported using TfL) vs a comparison group who should have limited exposure to the TfL policy. The combination of both a traditional DID design (given the control group) and a counterfactual design is confusing given that the results focused on the counterfactual approach. The authors need to be a better job explaining their choices. The study is an important one and can add significantly to the literature around unhealthy food marketing and regulations that need to work in concert to minimize it.

Specific Comments

1) Abstract Line 45: should be "energy and nutrients".

2) It might have been informative for the study to also have included a non-HFSS food category as a potential control food category to see if those trends were parallel for Londoners vs North Englanders. Alternatively it would have been useful to see what energy purchased from non-HFSS categories might have changed (or not).

3) Why didn't the authors consider using Feb 2018-Nov 2018 as the "pre" period and Feb 2019-Nov 2019 as the "post" period? Is there reason to think that the 3 months between the announcement and implementation would have had some impact on the outcomes? This also helps deal with potential seasonality issues. Or was the choice to only start later in 2018 related to wanting to avoid the SDIL implementation? If so be explicit.

4) It might be useful to provide a map to illustrate the boundaries and areas of residence among the sample used for the "treatment" vs " control" households.

5) Pls provide more information on how the authors determined the NPM scoring, particularly around FV content. What approximations did they have to use given that this information is not easily available?

6) The authors state that they were unable to assess volume purchased. How did then they obtain information about the total energy and nutrients purchased from the HFSS categories? Was the nutrient information available already able to account for volume? Typically the nutrient information is in the form of nutrient densities (per 100g or per 100ml), which then allows for "copying" of nutrient density values across products (adjusting for volume or package sizes), as well as for the application of the NPM. As such, it is unclear how the nutrient values are derived and why there aren't also volume measures, especially if package sizes can vary considerably.

7) Was the nutrient information updated within the time period of the data used in the analyses? Or was it the same throughout the period? It would be useful to be explicit about this because then it provide an indication as to whether some potential reformulation may have happened (or not).

8) Results on food category outcomes. The direction of the differences are unclear. For example, energy & sugar amounts from Sugary drinks, Sugary Cereals & Savory snacks were higher (not significantly) post-policy compared to the counterfactual, but the text does not make this clear and in fact the text. 

9) If the data/analyses already included a control group/sample of North Englanders, why not just do a DID analyses and not bother with a controlled ITS model with the construction of counterfactuals? It is unclear to me if the inclusion of the control sample in the model biases the estimates. Did the authors compare the results from a straightforward DID with covariates vs using a counterfactual? 

10) Figure 2 is rather blurry and the quality needs to be improved.

11) Was the study adequately powered to detect statistical significance for all main outcomes? Some indication of this would be useful.

12) Given the many outcomes, did the authors consider correcting for this? Bonferroni or Benjamini corrections may be needed.

13) Might the authors say anything about how a longer period under the TfL policy might have a bigger impact given the 10 month post- policy data used?? Would an event study design be possible to address this question?

14) Some version of Supplementary table 10 should be included in the main text as those values are useful to have as reference easily.

15) Can the authors say anything about what is known with regards to compliance with the policy? How was it being enforced? What were the penalties?

16) The analysis does a pre-post and is unable to delve into the mechanisms or monitor actual exposure to unhealthy food marketing (much of which is likely subliminal) and thus is limited to behavioral measures. Please add a short discussion around this and the need for research to better understand the mechanistic /theory of change.

***

[LINK]

---

## [Decision Letter · Decision Letter 2]

11 Jan 2022

Dear Dr. Yau,

Thank you very much for re-submitting your manuscript "Changes in household food and drink purchases following restrictions on the advertisement of high fat, salt and sugar products across the Transport for London network: A controlled interrupted time series analysis" (PMEDICINE-D-21-04172R2) for consideration at PLOS Medicine.

I have discussed the paper with our academic editor and it was also seen again by three reviewers. I am pleased to tell you that, provided the remaining editorial and production issues are fully dealt with, we expect to be able to accept the paper for publication in the journal.

[LINK]

Please let me know if you have any questions in the meantime, and we look forward to receiving the revised manuscript.   

Kind regards,

Richard Turner, PhD

rturner@plos.org

Requests from Editors:

At line 52 (abstract) and elsewhere we suggest avoiding the implied double negative ("-19.4% ... lower", in favour of "19.4% ... lower"; or "-19.4% change in ..."). 

At line 56, please adapt the language to avoid implying causality: "... so the policy was associated with attenuated growth ... rather than with reduced purchases ...". Please make similar changes throughout the paper where needed. 

At line 553, please make that "... possible longer-term impacts".

At line 560 and any other instances, please make that "... apparent effect sizes".

At line 569, please soften the language used: "... we observed a relative reduction ..." or similar.

Noting reference 32, please ensure that all citations have full access details. 

Comments from Reviewers:

*** Reviewer #1:

 Thanks for the revised manuscript and responses to my original queries. Overall, I am happy with the updates and recommend the manuscript be accepted. 

One small change - I do not think that incomplete household data is a big problem for this study - but if this was missing not-at-random it could change the findings (i.e. households not reporting data on weeks with a 'treat-heavy' shop). A simple acknowledgement of this as a limitation should be fine.

I agree with the authors about ITS vs DID (in the response to reviewers of the first version of the MS) - the advantage of ITS is that it's possible to directly quantify any slope (trend) changes (and overcome several biases) which gives much more information than a simple (adjusted) before-after difference. 

Figure 1 helps understand the control locations, and Figure 2 is much easier to read now with adjusted line widths. Sub-group analysis looks fine now.

*** Reviewer #2: 

All comments have been adequately addressed 

*** Reviewer #3: 

From what I can tell, the authors have adequately addressed the comments raised by the reviewers. This is an important paper and it would be good to have this published.

***

[LINK]

---

## [Editor Report · Decision Letter 3]

14 Jan 2022

Dear Dr Yau, 

On behalf of my colleagues and the Academic Editor, Dr Popkin, I am pleased to inform you that we have agreed to publish your manuscript "Changes in household food and drink purchases following restrictions on the advertisement of high fat, salt and sugar products across the Transport for London network: A controlled interrupted time series analysis" (PMEDICINE-D-21-04172R3) in PLOS Medicine.

Prior to final acceptance, please amend the text at line 141 (e.g. "Research ... has been published").

PRESS

Sincerely, 

Richard Turner, PhD 

rturner@plos.org